# Early Adoption of Checkpoint Inhibitors in Patients with Metastatic Gastric Adenocarcinoma—A Case Series of Non-Operative Long-Term Survivors

**DOI:** 10.3390/diseases10020024

**Published:** 2022-04-24

**Authors:** Dalia Kaakour, Garrett Ward, Farshid Dayyani

**Affiliations:** 1Department of Medicine, University of California, Irvine, CA 92697, USA; dkaakour@hs.uci.edu; 2Department of Radiology, University of California, Irvine, CA 92697, USA; ggward@hs.uci.edu; 3Division of Hematology and Oncology, Department of Medicine, University of California, Irvine, CA 92697, USA

**Keywords:** checkpoint inhibitors, advanced gastric cancer, survival, durable remission

## Abstract

Checkpoint inhibitor (CPI) therapy has only recently been introduced in the first-line treatment of advanced gastric cancer. However, later line monotherapy CPI efficacy in a subset of patients was presented about four years prior. Here, we present three cases of advanced gastric adenocarcinoma cancers treated with CPI in early lines years prior to the availability of randomized first line data. All three patients remain in remission without gastrectomy, with the median time from initial diagnosis of approximately 52 months. With long-term follow-up of more than four years, we present a proof of concept that, with early integration of CPI therapy, highly durable responses are possible even in the absence of surgery in patients with advanced gastric and gastroesophageal junction cancers.

## 1. Introduction

As of 2020, the global annual incidence of gastric cancer is about 1.09 million new cases per year, with about 769,000 deaths per year [1]. Long-term survival in the vast majority of cases is only achieved in patients with locoregional disease and involves gastrectomy. However, gastric cancers are often diagnosed at advanced stages, and survival rates remain poor despite advances in systemic treatments. Checkpoint inhibitors (CPIs) have shown promising outcomes in a variety of advanced malignancies outside of gastrointestinal malignancies such as in melanoma, bladder and lung cancers [2]. In the United States, in 2021, pembrolizumab (an anti-PD-1 antibody) was approved for first-line therapy in patients with HER2 positive disease, combined with trastuzumab and chemotherapy [3]. The only other CPI approved for first-line therapy in advanced gastric cancer is nivolumab (also an anti-PD-1 antibody) in combination with chemotherapy for patients with a CPS score greater than or equal to five in the treatment of advanced or metastatic gastric cancer, gastroesophageal junction cancer and esophageal adenocarcinoma based on CheckMate 649 (April 2021) [4].

Both approvals are very recent, and prior to those, CPIs were not included in the treatment guidelines for first-line treatments in advanced gastric cancer. However, early data on the efficacy of single-agent pembrolizumab in a subset of patients with refractory gastric cancer (especially in those with PD-L1-positive tumors) was already presented in 2017 (KEYNOTE-059) [5]. Based on the observation of durable long-term responses seen in KEYNOTE-059 and subsequent trials in gastric cancer, we adopted the inclusion of CPI agents (with cytotoxic therapy) early on in patients with unresectable gastric cancer and PD-L1-positive tumors. Here, we present three patients diagnosed with stage IV gastric cancer and alive four years or more after first-line treatment with CPI containing regimens.

## 2. Case Series

Three patients with metastatic poorly differentiated gastric adenocarcinoma who responded to chemotherapy initially and who then discontinued chemotherapy due to either drug toxicity (case 1 and 2) or due to disease progression (case 3) were all subsequently treated with monotherapy on checkpoint inhibitors. Metastatic disease was diagnosed per guidelines using endoscopic ultrasound and PET CT (case 1 and 2) or CT plus liver biopsy (case 3). All scans were interpreted by fellowship-trained academic abdominal radiologists. All patients developed an initial documented benefit on pembrolizumab or nivolumab, with two patients eventually showing no evidence of disease and the third showing continued partial response. All three patients are still alive and continue to show the aforementioned responses to date (March 2022). Patient characteristics, cancer type and location, and treatment information are shown in Table 1.

Case 1 represents a patient that was treated with single-agent 200 mg IV pembrolizumab every three weeks for a total of 23 cycles (21 months), when treatment was stopped due to patient request. Patient was lost to follow-up intermittently but was seen again recently and found to be clinically stable with no known evidence of disease to date. Case 2 represents a patient who was on a course of single-agent 480 mg IV nivolumab every three weeks for a total of 6 cycles (4.5 months), was then transitioned to single-agent 200 mg IV pembrolizumab every three weeks for 26 cycles (18 months) and was then transitioned to single-agent 400 mg IV pembrolizumab every six weeks for 16 cycles (21 months). Case 3 represents a patient who was on a course of single-agent 200 mg IV pembrolizumab every three weeks for 16 cycles (18.5 months) and was then transitioned to single-agent 400 mg IV pembrolizumab every six weeks for 11 cycles (20.5 months).

### 2.1. Case 1

Case 1 is a patient with poorly differentiated gastric adenocarcinoma with peritoneal carcinomatosis. The patient was initially treated with docetaxel, oxaliplatin, leucovorin and 5-fluorouracil (FLOT) for two complete cycles until oxaliplatin was held for elevated LFTs and thrombocytopenia. The patient had a marked response to this treatment after three months. However, due to worsening thrombocytopenia, he was switched to pembrolizumab. Follow-up EGD showed negative biopsies, and surveillance imaging since this time has shown continued improvement with no measurable disease (Figure 1, left panel). The patient stopped treatment after cycle 23 due to wanting a treatment break. As of the time of case series writing, he remains living for 56 months since initial diagnosis.

### 2.2. Case 2

Case 2 is a patient with poorly differentiated gastric adenocarcinoma with intrathoracic and intraabdominal nodal metastases. The patient was initially treated with folinic acid, fluorouracil and oxaliplatin (FOLFOX), which he was responding to; however, he was then switched to nivolumab due to anorexia. Surveillance imaging three months after initiating nivolumab showed improvement in the primary mass, with EGD/EUS showing multiple malignant nodes in the mediastinum. By about 5.5 months after initiating nivolumab, surveillance imaging showed no evidence of metastatic gastric adenocarcinoma within the abdomen or pelvis. At this time, the patient was switched to pembrolizumab due to an extended treatment cycle of six weeks, and follow-up imaging showed a resolution of the gastric body mass. Multiple FDG-avid abdominal lymph nodes were resolved, and persistent FDG-avid mediastinal bilateral hilar lymph nodes were present. Eventually, PET CT showed slight interval decrease in the FDG activity of mediastinal and hilar lymph nodes (Figure 1, middle panel). The patient continues on treatment with pembrolizumab every six weeks at this time. As of the time of case series writing, he remains living for 52 months since initial diagnosis.

### 2.3. Case 3

Case 3 is a patient with poorly differentiated gastric adenocarcinoma with innumerable hepatic lesions. The patient was initially treated with docetaxel, oxaliplatin, leucovorin and 5-fluorouracil (FLOT) chemotherapy, and after three months of therapy, repeat imaging showed no significant change in the primary gastric antral region or in the numerous hepatic metastases. The patient was then initiated on pembrolizumab, and four days after the start of cycle 1 pembrolizumab, he received Trans-arterial Tirapazamine Embolization (TATE) to the liver. Follow-up CT scan three months after showed liver lesions slightly smaller than before, and imaging four months after this showed interval increase in size and number of multiple hepatic metastases. The patient went on to undergo TATE to the liver again along with tirapazamine (TPZ) to left the hepatic lobe and was treated with paclitaxel and ramucirumab, to which he showed partial response in the liver. Eventually, paclitaxel was stopped due to neuropathy and ramucirumab was stopped due to insurance denial. The patient continued on pembrolizumab, has shown significant interval improvement of multiple hepatic lesions and shows continued response to date (Figure 1, right panel). The patient continues on treatment with pembrolizumab every six weeks at this time. As of the time of case series, he remains living for 47 months since initial diagnosis.

## 3. Discussion

Although monotherapy with checkpoint inhibitors as the second line in the treatment of advanced gastric and gastroesophageal junction cancer has yet to be guideline approved, in 2017, the data demonstrated CPI activity used in later line therapy in these cancers. The first two notable studies include the KEYNOTE-059 study using pembrolizumab (in Asia, the USA and other parts of the world) and the ATTRACTION-2 study using nivolumab (in Japan, South Korea and Taiwan) [5,6].

In the KEYNOTE-059 Trial, a phase 2 trial that evaluated the safety and efficacy of 200 mg IV pembrolizumab every three weeks in a cohort of patients with previously treated gastric or gastroesophageal junction cancer, a subset analysis showed that PD-L1 expression enhanced response to pembrolizumab plus chemotherapy. Additionally, a subset of patients had durable long-term responses. However, the response rate with a single agent was just 11.6% (95% CI 8.0–16.1). Pembrolizumab was tolerated well both as a monotherapy and in combination with chemotherapy by patients in this study [5].

Moving forward, the question of whether CPI therapy could be introduced earlier in the course of treatment in patients with advanced gastric and gastroesophageal junction cancer led to further studies. The ATTRACTION-2 study was the first phase 3 study to examine patients with gastric and gastroesophageal junction cancer who had previously been treated with two or more chemotherapy agents, now treated with nivolumab monotherapy. When looking at the trial’s two-year updated data, there was a higher overall survival (OS) rate in the nivolumab vs. placebo group both at one year and at two years (27.3% vs. 11.6% and 10.6% vs. 3.2%, respectively). The OS benefit was observed regardless of tumor PD-L1 expression. In the nivolumab group, among patients with a complete or partial response, the median OS rates were 87.1% at one year and 61.3% at two years [6].

KEYNOTE-061 then tested monotherapy with pembrolizumab for up to two years versus standard-dose paclitaxel in the second line for patients with advanced gastric and gastroesophageal junction cancer. This trial confirmed that patients with a high CPS score (PD-L1 CPS of 1 or higher) benefit from pembrolizumab and can have durable responses, with a median overall survival of 9.1 months with pembrolizumab and 8.3 months with paclitaxel. However, median progression-free survival was 1.5 months with pembrolizumab and 4.1 months with paclitaxel [7].

To date, three trials in the first-line setting have reported results for CPI added to chemotherapy: CHECKMATE-649, ATTRACTION-4 and KEYNOTE-811. The CHECKMATE-649 trial was a phase 3 trial evaluating nivolumab plus ipilimumab and nivolumab and chemotherapy verses chemotherapy alone as a first-line treatment in patients with advanced gastric and gastroesophageal junction cancer [4]. Similarly, the ATTRACTION-4 study examined the safety and efficacy of nivolumab combined with S-1 plus oxaliplatin or capecitabine plus oxaliplatin as a first-line therapy for unresectable advanced or recurrent human epidermal growth factor receptor 2 (HER2)-negative gastric and gastroesophageal junction cancer in Asian populations [8]. The KEYNOTE-811 trial looked at pembrolizumab or placebo plus trastuzumab plus chemotherapy in human epidermal growth factor receptor 2 positive (HER2+) advanced gastric and gastroesophageal junction cancer [3]. All three of these trials showed improved outcomes with the addition of CPI to chemotherapy, although OS was not statistically improved in ATTRACTION-4.

Here, we reported a case series more specifically looking at patients with advanced gastric adenocarcinomas with high CPS scores (10–50%) (Table 2) who did not undergo surgery, who were treated early with CPI as a means of maintenance after induction chemotherapy, and either showed no evidence of disease or a very durable continued partial response. 

Traditionally, a durable remission has not been possible without surgical resection. We hypothesize that early integration of immunotherapy without surgery in patients with advanced gastric and gastroesophageal junction cancers may lead to durable remissions. Thus far, the median time from initial diagnosis to date for these patients is approximately 52 months and ongoing. These patients are approaching the definition of cured (five years) without surgery. It is unclear at which treatment length should immunotherapy continue on for, as Patient #1 stopped treatment at cycle 23 and remains with no evidence of disease, while Patients #2 and #3 continue on CPI treatment. Patient #3 uniquely highlights the possibility of immune activation via liver directed therapy, as he initially benefitted from pembrolizumab and then had disease progression; however, after liver directed therapy, he resumed single-agent pembrolizumab and continues to see a sustained response. Most contemporary CPI containing trials stop treatment after two years without progression. Thus, the two-year mark might serve as a guide for when to discuss a treatment break with patients in remission. According to the National Cancer Institute, exceptional responders are defined as “patients who have dramatic and long-lasting responses to treatments for cancer that were not effective for most similar patients”. Based on this definition and using the most recent Phase 3 first line data for chemotherapy plus CPI (i.e., Checkmate 649 study), we would expect a median PFS of 7.7 months and a median OS of 13.8 months. Since our patients have achieved a complete response in two of three cases and exceeded the median PFS and OS by more than 300%, it would be appropriate to categorize them as exceptional responders. Of note, none of the patients underwent radiation therapy. Hence, it would not be possible to speculate on the potential additive abscopal effect of radiation in our patients.

In summary, we present a series that highlights that careful examination of available data and rational extrapolation might lead to early benefit for patients, even in the absence of mature phase 3 data. Reliable predictive biomarkers for long-term remission after CPI containing regimens are needed. Additionally, the optimal duration or choice of the induction phase with chemotherapy is yet to be determined. As seen in the Keynote-062 study, continuation of both chemotherapy plus pembrolizumab until progression was actually not superior to chemotherapy alone despite a higher response rate [9]. This finding suggests that prolonged exposure to cytotoxic therapy might lead to loss of immune response. The patients treated in our series underwent about three months of cytotoxic therapy before a switch to single-agent CPI. Interestingly, patient #3 had progression after 1L pembrolizumab and was enrolled on a protocol of pembrolizumab plus liver embolization with tirapazamine. Rechallenging the patient with pembrolizumab has led to a durable, at least partial response. Although the liver lesions are still present on imaging, repeat liver biopsy has not been performed to test whether the lesions are necrotic. A follow-up trial is further testing this approach is currently ongoing (NCT04701476). Longer-term follow up of the completed 1L trials with chemotherapy and IO and their subset analyses are expected to help with better patient selection in the future.

## Figures and Tables

**Figure 1 diseases-10-00024-f001:**
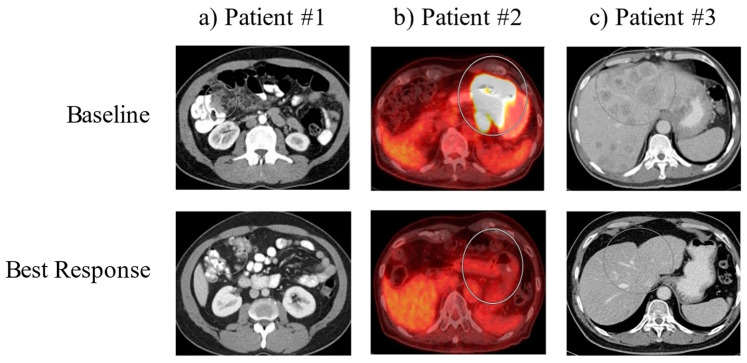
Baseline and best response imaging. Representative images of the tumors for each patient (in circles) are shown at the time of diagnosis (Upper Panel: ‘Baseline’) and at the time of maximum tumor shrinkage (Lower Panel: ‘Best Response’).

**Table 1 diseases-10-00024-t001:** Patient characteristics, cancer type and treatment.

Case	Age (Years)	Gender	Primary	Site of Metastasis	Agent Type	Number of Cycles	Months of CPI	Response
1	59	Male	poorly diff gastric adenocarcinoma	peritoneal carcinomatosis	pembrolizumab	23	21	No evidence of Disease
2	79	Male	poorly diff gastric adenocarcinoma	intrathoracic and intraabdominal nodal metastases	nivolumab pembrolizumab	6 26 (200 mg) + 16 (400 mg) = 42	4.5 18 + 21 = 39	No evidence of Disease
3	61	Male	poorly diff gastric adenocarcinoma	innumerable hepatic lesions	pembrolizumab	16 (200 mg) + 11 (400 mg) = 27	18.5 + 20.5 = 39	Continued partial response

**Table 2 diseases-10-00024-t002:** Molecular profiling.

Case	HER2	IHC	MSI	PDL1 (CPS)	NGS
1	negative	Strongly + CK7 +CDX2 (40%) Negative CK20 Negative H pylori	Cannot be determined	+(35%)	ATR L1707I PIK3R1 T576del TMB cannot be determined
2	negative	Negative CK7 +CDX2 Negative CK20 Positive AE1/AE3 Positive H pylori	MS-stable	+(50%)	MET amplification CDK6 amplification—equivocal EP300 S1476fs*1 FUBP1 I436fs*7 MAP2K4 loss TP53 E287*
3	negative	Patchy + CK7 Diffuse positive CDX2 Negative CK20 Weak rare positive HCC Negative SALL4 PAS positive PAS-D negative	MS-stable	+(10%)	PTEN N340fs 0.4% Temsirolimus, Copanlisib, Everolimus TP53 P316fs 0.9% Negative for MUC17 or CLDN18.2

HER2: human epidermal growth factor receptor 2; CK: cytokeratin; IHC: immunohistochemistry; MSI: microsatellite instability; PD1: Programmed death ligand-1; CPS: combined positive score; NGS: Next Generation Sequencing; TMB: Tumor Mutations Burden.

## Data Availability

Study does not include data.

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
