# Peer review of "Early Adoption of Checkpoint Inhibitors in Patients with Metastatic Gastric Adenocarcinoma—A Case Series of Non-Operative Long-Term Survivors"

_diseases, 2022, doi:10.3390/diseases10020024_

Round 1

Reviewer 1 Report

This is a very interesting and original report. However we need to have on the case report the pathological analysis confirming the metastatic process, as liver biopsy or peritoneal biopsy.

This is important to convince.

For the rest – good paper

Author Response

Dear Editor,

We would like to extend our gratitude to both reviewers for their insightful and prompt comments.

Per the institutional review board (IRB) guidelines of the University of California Irvine, case reports and small case series do not constitute research and thus are IRB exempt.
In addition, the complete de-identification of the subject presentations and lack of inclusion of any personal information means a written informed consent is not mandatory.
We have obtained verbal consent from all three subjects, but since all of them are on long-term follow-up with infrequent clinic visits, it is not feasible (and based on the above guidelines not necessary) to obtain written consents in a timely manner.

Please find below a point by point response to the reviewer comments:

Reviewer #1:
This is a very interesting and original report. However we need to have on the case report the pathological analysis confirming the metastatic process, as liver biopsy or peritoneal biopsy.
This is important to convince.

Author Response:
Thank you very much for this suggestion. Indeed, all patients underwent at initial diagnosis thorough diagnostic evaluations to establish the disease stage, including gastric biopsy to establish diagnosis, as well as endoscopic ultrasound and either CT scans (with and without contrast) and or PET-CT scans, as recommended by NCCN guidelines.
Where medically feasible and safe, biopsy of the affected organ was performed. This was the case for patient #3 who had liver metastases and underwent embolization of those lesions.
However, patients #1 and #2 both had advanced disease based on positive diagnostic PET CT scans read by experienced academic radiologists with abdominal subspecialization. 
In both cases, based on the location (diffuse peritoneal nodules and intra-throacic/intra-abdominal adenopathy, respectively) it was not deemed medically safe to obtain tissue biopsy, given the clear clinical and PET CT findings. 
All patients underwent ctDNA testing via liquid biopsies.

Sincerely,

Drs. Dalia Kaakour and Farshid Dayyani on behalf of all the authors

Reviewer 2 Report

 Thank you for your thoughtful paper about non-operative long survivor treated by CPI. The patients achieved good response had valuable clinical course.  I have some question.

  1. We experienced the same case with achieved good response without surgery. However, it is difficult to find the difference of non-response cases. Further study is required to investigate the sensitivity. Please discuss the difference between good response and non-response cases clearly if possible.
  2. Please mention radiation therapy and CPI, not only the correlation between CPI and chemotherapy.

Author Response

Dear Editor,

We would like to extend our gratitude to both reviewers for their insightful and prompt comments.

Per the institutional review board (IRB) guidelines of the University of California Irvine, case reports and small case series do not constitute research and thus are IRB exempt.
In addition, the complete de-identification of the subject presentations and lack of inclusion of any personal information means a written informed consent is not mandatory.
We have obtained verbal consent from all three subjects, but since all of them are on long-term follow-up with infrequent clinic visits, it is not feasible (and based on the above guidelines not necessary) to obtain written consents in a timely manner.

Please find below a point by point response to the reviewer comments:

Reviewer #2:
1.    We experienced the same case with achieved good response without surgery. However, it is difficult to find the difference of non-response cases. Further study is required to investigate the sensitivity. Please discuss the difference between good response and non-response cases clearly if possible.

Author Response: This is a very important point.

The NCI defines exceptional responders as "patients who have dramatic and long-lasting responses to treatments for cancer that were not effective for most similar patients".
Based on this definition, and using the most recent Phase 3 first line data for chemo+CPI (i.e. Checkmate 649 study), we would expect a median PFS of 7.7 months and a median OS 13.8 months.
Since our patients have achieved a complete response in 2/3 cases and exceeded the median PFS and OS by > 300%, it would be appropriate to categorize them as exceptional responders.

2.    Please mention radiation therapy and CPI, not only the correlation between CPI and chemotherapy.

Author Response: We fully agree with the reviewer that this is a very interesting area of research. However, none of our patients received radiation therapy, hence any discussion regarding radiation in the context of our case series would be speculative.
We look forward to reading reports from other investigators who might have more experience with XRT and CPI combinations.

Sincerely,

Drs. Dalia Kaakour and Farshid Dayyani on behalf of all the authors